# Palliative Sedation in COVID-19 End-of-Life Care. Retrospective Cohort Study

**DOI:** 10.3390/medicina57090873

**Published:** 2021-08-26

**Authors:** Jose-Manuel Ramos-Rincon, Oscar Moreno-Perez, Nazaret Gomez-Martinez, Manuel Priego-Valladares, Eduardo Climent-Grana, Ana Marti-Pastor, Joaquin Portilla-Sogorb, Rosario Sanchez-Martinez, Esperanza Merino

**Affiliations:** 1Internal Medicine Department, Alicante General University Hospital—Alicante Institute of Health and Biomedical Research (ISABIAL), 03010 Alicante, Spain; anamartipastor@gmail.com (A.M.-P.); portilla_joa@gva.es (J.P.-S.); sanchez_rosmar@gva.es (R.S.-M.); 2Clinical Medicine Department, Miguel Hernández University, 03550 Elche, Spain; omorenoperez@hotmail.es; 3Endocrinology and Nutrition Department, Alicante General University Hospital—Alicante Institute of Health and Biomedical Research (ISABIAL), 03010 Alicante, Spain; nazaret.gomez@goumh.umh.es; 4Palliative Care Unit and Internal Medicine Department, Alicante General University Hospital—Alicante Institute of Health and Biomedical Research (ISABIAL), 03010 Alicante, Spain; priego_man@gva.es; 5Pharmacy Department, Alicante General University Hospital—Alicante Institute of Health and Biomedical Research (ISABIAL), 03010 Alicante, Spain; climent_edu@gva.es; 6Unit of Infectious Diseases, Alicante General University Hospital—Alicante Institute of Health and Biomedical Research (ISABIAL), 03010 Alicante, Spain; merino_luc@gva.es

**Keywords:** COVID-19, palliative care, palliative medicine, death, symptom assessment

## Abstract

*Background and Objectives:* Descriptions of end-of-life in COVID-19 are limited to small cross-sectional studies. We aimed to assess end-of-life care in inpatients with COVID-19 at Alicante General University Hospital (ALC) and compare differences according to palliative and non-palliative sedation. *Material and Methods*: This was a retrospective cohort study in inpatients included in the ALC COVID-19 Registry (PCR-RT or antigen-confirmed cases) who died during conventional admission from 1 March to 15 December 2020. We evaluated differences among deceased cases according to administration of palliative sedation. *Results:* Of 747 patients evaluated, 101 died (13.5%). Sixty-eight (67.3%) died in acute medical wards, and 30 (44.1%) received palliative sedation. The median age of patients with palliative sedation was 85 years; 44% were women, and 30% of cases were nosocomial. Patients with nosocomial acquisition received more palliative sedation than those infected in the community (81.8% [9/11] vs 36.8% [21/57], *p* = 0.006), and patients admitted with an altered mental state received it less (20% [6/23] vs. 53.3% [24/45], *p* = 0.032). The median time from admission to starting palliative sedation was 8.5 days (interquartile range [IQR] 3.0–14.5). The main symptoms leading to palliative sedation were dyspnea at rest (90%), pain (60%), and delirium/agitation (36.7%). The median time from palliative sedation to death was 21.8 h (IQR 10.4–41.1). Morphine was used in all palliative sedation perfusions: the main regimen was morphine + hyoscine butyl bromide + midazolam (43.3%). *Conclusions*: End-of-life palliative sedation in patients with COVID-19 was initiated quite late. Clinicians should anticipate the need for palliative sedation in these patients and recognize the breathlessness, pain, and agitation/delirium that foreshadow death.

## 1. Introduction

The COVID-19 pandemic has entailed more than 181 million known infections and well over 3.92 million deaths globally as of 29 June 2021 (https://coronavirus.jhu.edu/map.html), posing enormous challenges for health systems worldwide, which have assumed a great burden of care. COVID-19 has a high mortality rate in patients requiring hospitalization, especially in older patients [1,2,3], prompting reflection on the need for palliative care in this population [3]. To date, studies have analyzed the support and involvement of the palliative medicine unit in end-of-life care in COVID-19 [4,5], the experience of palliative departments and hospital palliative care with these patients [6,7,8], the management of COVID-19 end-of-life care by clinicians who are not specialists in palliative care [9], and the changes that COVID-19 has brought about in the care of oncological and non-oncological palliative patients [10,11,12]. 

According to the WHO, palliative care is defined as “an approach that improves the quality of life of patients and their families facing the problem associated with life-threatening illness, through the prevention and relief of suffering by means of early identification and impeccable assessment and treatment of pain and other problems, physical, psychosocial and spiritual”. In other words, palliative sedation is a part of palliative care (alleviating refractory symptoms prior to death). It consists of inducing a state of decreased or absent awareness (unconsciousness) and relieving the burden of otherwise intractable suffering until death. The use of palliative sedation is an indicator of the quality of end-of-life care [8].

There are symptoms that are common at the end of life as COVID-19 disease progresses. These symptoms, such as dyspnea or delirium can become distressing and even intolerable for patients. The use of palliative sedation prior to death in patients with COVID-19 has been described in small cohorts [9,13,14]. We were interested in analyzing the clinical–epidemiological characteristics of patients who died from COVID-19 in the medical ward during the first and second wave of infections and compare patients who received or did not receive palliative sedation. We also describe the main characteristics of palliative sedation.

## 2. Materials and Methods

Design: This was a retrospective cohort study in consecutive inpatients who died of COVID-19 in a Spanish tertiary hospital in metropolitan Alicante, located on the Mediterranean coast of Spain. 

Participants: We included patients of all ages who died from 1 March to 15 December 2020 due to COVID-19, confirmed by RT-PCR or antigen testing. The first wave was defined as the period from 1 March to 30 of June and the second from 1 July to 15 December 2020. 

Data collection: Manual chart reviews were complemented by data extraction from electronic health records (EHRs). Trained physicians collected exploratory variables such as demographics, known SARS-CoV-2 epidemiological risk factors, comorbid conditions, medications, COVID-19 related symptoms, laboratory tests, and outcomes. The main explanatory variable was the palliative sedation of patients’ death, defined as the use of sedatives to reduce the patient’s level of consciousness and alleviate refractory symptoms before death, that is, continuous deep sedation [14]. The hospital had a protocol for managing COVID-19 patients admitted in the hospital with specific criteria for the use of palliative sedation by a palliative care team in patients with clinically worsening conditions, refractory symptoms, and/or a short-term fatal prognosis. The reasons for palliative sedation were decided by the patient’s physician according to this protocol.

Statistical analysis: Categorical and continuous variables are given as frequencies (percentages) and as medians (interquartile range, IQR), respectively. We compared palliative sedation and non-palliative sedation using the Mann–Whitney U for numeric traits and Chi-squared and Fisher’s exact tests for binary outcomes, as appropriate. All tests were two-sided, and *p* values under 0.05 were considered statistically significant. IBM SPSS Statistics v25 (Armonk, NY, USA) was used for analyses. 

Ethical aspects: The study was conducted in accordance with the Declaration of Helsinki, and the protocol was approved by the Alicante General University Ethics Committee (ID code number: 200145, date of approved: 1 June 2020). All patients gave their informed consent for inclusion before they participated in the study from 1 June. For the remaining patients, included retrospectively, informed consent was waived. The STROBE statement guidelines were followed in the conduct and reporting of the study.

## 3. Results

Of 747 patients hospitalized in our center with COVID-19 in the study period, 101 (13.5%) died (49/301 [16.3%] during the first wave and 52/446 [11.7%] during the second; *p* = 0.117). Sixty-eight (67.3%) died in acute medical hospitalization and 33 (32.7%) in the intensive care unit (ICU). Of the inpatients in conventional wards, 30 (44.1%) received palliative sedation (38.5% in wave 1 vs 50.0%, in wave 2).

Table 1 shows demographic characteristics, comorbidities and clinical presentation according to palliative sedation. The median age of patients receiving this care was 85 years (IQR 77–90); 44% were women, and 16.2% of cases were nosocomial COVID-19. The median length of stay in patients with palliative sedation was 9 days (IQR 5–12), slightly more than patients without palliative sedation (5 days, IQR 2–13). Patients with nosocomial acquisition received more palliative sedation (81.8%, [9/11]) than those infected in the community or care homes (36.8% [21/57], *p* = 0.006). Patients admitted with an altered mental state received it less (20% [6/23] vs. 53.3% [24/45], *p* = 0.032).

Groups with and without palliative sedation were similar in terms of median time from symptoms onset to hospitalization; the presence of fever, respiratory and systemic symptoms on initial presentation; and steroid and tocilizumab treatment. Patients admitted with an altered mental state received less palliative sedation (20% [6/23] vs. 53.3% [24/45], *p* = 0.032). 

Table 2 shows the characteristics of palliative sedation use. The median time from admission to initiating palliative sedation was 8.5 days (IQR 3.0–14.5). The indication for palliative sedation was recorded in the EHR, and patients’ relatives were informed. The main symptoms prompting palliative sedation were dyspnea at rest (90%), pain (60%), and agitation/delirium (36.7%). Palliative sedation was administered intravenously; active treatment was withdrawn in 96.7% of patients, the same proportion for which no hydration was performed. The median duration of sedation was 21.8 h (IQR 10.4–41.1) until death. Morphine was used in all palliative sedation perfusions, and the main prescribed regimen was morphine + hyoscine butyl bromide + midazolam (43.3%). All patients received oxygen therapy for comfort.

## 4. Discussion

Just under half the patients who died from COVID-19 received palliative sedation in acute medical wards. Palliative sedation, prescribed to induce a state of decreased or absent awareness (unconsciousness) and relieve the burden of otherwise intractable suffering until death, was generally administered for fewer than 24 h by the attending general physician rather than by the attending palliative physician. In this study, palliative sedation was defined solely by the use of drugs to reduce the level of consciousness when there was no response to intensive symptom-control treatment. These were considered refractory symptoms due to the patients’ end-of-life situation.

In this study, the increased sedation administered to nosocomial COVID-19 cases is probably due to longer admissions and more comorbidities, making symptom control more relevant. The lower levels of palliative sedation in patients with an altered of level of consciousness may stem from the presence of dementia, which is known to be associated with less palliative sedation [15].

The most commonly reported symptoms in the final days of life were dyspnea at rest, pain, and delirium/agitation, as reported elsewhere [4,5,6,8,12,13]. Cough is often a presenting symptom of COVID-19 but was not an ongoing issue in our cohort, unlike others [12].

The clinicians who prescribed palliative sedation in our patients were on-call or regular staff, not palliative care specialists. Palliative sedation was used for short periods of time (median 21.8 h), less than in other studies where medical staff or advising physicians from palliative care units prescribed the palliative sedation for a longer period for symptom management [6,8].

A recent systematic review of the clinical aspects of palliative sedation in the non-COVID-19 population showed that the decision-making process was led mainly by the palliative care team, and in some cases by the attending palliative care physician [16]. Another study, focused on general practitioners, highlighted the importance of working with palliative care professionals due to GPs’ limited experience in palliative sedation [17]. In the case of COVID-19, an acute infectious disease with high mortality, several hospitals have developed guidelines that could be used for attending physicians who are not specialists in palliative care, with objective criteria for palliative care sedation in these situations [18]. 

The main medications used were morphine and midazolam, in line with previous studies [4,5,6,8,12,13]. Other opioids like alfentanil, fentanyl, buprenorphine, hydromorphone, and oxycodone, associated with a finer approach to alleviating refractory symptoms prior to death, were not used. Other sedatives like diazepam, clonidine, clonazepam, and olanzapine were likewise not administered. Moreover, since we excluded patients who died in the ICU with invasive mechanical ventilation, we did not register the use of propofol or other anesthetics. The use of glycopyrrolate, for controlling respiratory secretions, was similar to other studies [6,8]. However, our patients received hyoscine for noisy respiratory secretion, a drug used little in the UK, Italy, and Australia [4,5,6,8,12,13].

This study has the limitations of any single-center retrospective study. Moreover, the sample size was small, and we did not analyze palliative sedation in patients admitted to the ICU. Additionally, this research only studied the palliative sedation in COVID-19 patients with advanced stage disease, but they received high-quality palliative care with particular attention paid to the bereavement management of the family/caregivers and in the right approach to psychological problems.

The main strength of this study is that it analyzes the experience of a public hospital where patients were attended by different medical and surgical specialists, with limited training in palliative care. These circumstances reinforce and validate our real-world results in the pandemic context.

## 5. Conclusions

In conclusion, end-of-life palliative sedation in patients with COVID-19 was initiated quite late. Clinicians should anticipate palliative sedation in these patients and recognize the breathlessness, pain, and agitation/delirium that foreshadow death. These results suggest the need to correctly identify refractory symptoms at the end of life in COVID-19 patients and integrate pharmacological sedation in the care for inpatients with COVID-19 in acute medical wards attended by physicians not specialized in palliative care. Palliative sedation must be a mandatory component of multidisciplinary end-of-life approach in COVID-19.

Future research is needed to evaluate the effectiveness of palliative sedation in COVID-19 patients considering the relief of the refractory symptom burden. There is also a need to advance our understanding of effective palliative sedation, including family and caregiver views in COVID-19 patients.

## Figures and Tables

**Table 1 medicina-57-00873-t001:** Demographic characteristics, comorbidities, and clinical presentation according to the administration of palliative sedation, in inpatients with COVID-19 in an acute medical ward.

	Palliative Sedation (*n* = 30)	No Palliative Sedation (*n* = 38)	*p* Value
**Demographics**			
Wave 1	15 (50.0)	24 (63.2)	0.28
Wave 2	15 (50.0)	16 (36.8)	0.28
Age in years, median (IQR)	85.0 (77–90)	88 (82–90)	0.43
Women, *n* (%)	13 (41.9)	18 (58.1)	0.74
Long-term care resident, *n* (%)	6 (20	10 (26.3)	0.54
Nosocomial, *n* (%)	9 (30.0)	2 (5.3)	**0.006**
**Comorbidities**			
Hypertension, *n* (%)	25 (83.3)	31 (81.6)	0.85
Diabetes mellitus, *n* (%)	16 (53.3)	17 (44.7)	0.48
Chronic respiratory disease, *n* (%)	10 (33.3)	14 (36.8)	0.77
Dementia, *n* (%)	8 (26.7)	16 (42.1)	0.12
Charlson comorbidity index, median (IQR)	7.5 (6.0–9.0)	7.0 (6.08.0)	0.54
10-year expected survival, median (IQR)	0.005 (0.00–3.6)	0.01 (0.00–2.5)	0.95
Clinical frailty scale ≥5, *n* (%)	24 (80.0)	25 (65.6)	0.30
**Clinical Presentation**			
Days from symptoms onset to hospitalization, median (IQR)	3.0 (2.0–7.0)	4.0 (2.0–7.0)	0.80
Fever, *n*/*N* (%)	15/29 (51.7)	16/36 (44.4)	0.56
Dry cough, *n*/*N* (%)	15/29 (51.7)	14/36 (38.9)	0.30
Wet cough, *n*/*N* (%)	3/29 (10.3)	6/36 (16.7)	0.46
Dyspnea, *n*/*N* (%)	25/30 (83.3)	29/38 (76.3)	0.48
Fatigue, *n*/*N* (%)	9/29 (31.0)	10/36 (27.8)	0.77
Myalgias-arthralgias, *n*/*N* (%)	3/29 (10.3)	1/37 (2.7)	0.20
Diarrhea, *n*/*N* (%)	3/29(10.3)	2/36 (5.6)	0.42
Altered mental state, *n*/*N* (%)	6/30 (20.0)	17/38 (44.7)	**0.032**
Initial Assessment			
Room-air pulse oximetry, median (IQR)	93 (90–97.0)	92.0 (86.0–97.0)	0.62
Systolic blood pressure, mmHg, median (IQR)	130 (120–150)	130 (115–143)	0.62
Respiratory rate, breaths/min, median (IQR)	27 (20–32)	22 (16–28)	0.22
**Imaging**			
Abnormal chest X-ray, *n* (%)	21 (70.0)	26 (68.4)	0.89
Opacities >50% of lung surface on X-ray, *n*/*N* (%)	8/21 (38.1)	12/26 (46.2)	0.58
**Treatment during admission**			
Tocilizumab, *n* (%)	6 (18.5)	6(15.8)	0.70
Corticosteroid, *n* (%)	20 (52.6)	18 (47.3)	0.12
**Complications**			
Pulmonary embolism, *n*/*N* (%)	2/28 (7.1)	0/37 (0.0)	0.13
**Death**			
Directly related with COVID-19, *n* (%)	23 (76.6)	28 (73.7)	0.61
Days from symptoms onset to death, median (IQR)	13.0 (6.0–18.5)	12.0 (9.0–20.5)	0.80
Length of hospital stay, days, median (IQR)	9.0 (5.0–12.0)	5 (2.0–13.0)	0.28

IQR: interquartile range. Statistically significant differences shown in bold.

**Table 2 medicina-57-00873-t002:** Characteristics of palliative sedation use to alleviate refractory symptoms prior to death in inpatients with COVID-19 in an acute medical ward (*n* = 30).

Variables		Values
**General**		
Recorded in electronic health records, *n* (%)		30 (100)
Consent requests, *n* (%)		30 (100)
Days from admission to starting palliative sedation, median (IQR)		8.5 (3.0–14.5)
Hours from initiation of palliative sedation to death, median (IQR)		21.8 (10.4–41.1)
**Symptoms of terminal illness, *n* (%)**		
Dyspnea at rest		27 (90.0)
Pain		18 (60.0)
Agitation/delirium		11 (36.7)
Vomiting		4 (13.3)
**Treatment, *n* (%)**		
Intravenous perfusion		27 (100)
Suspension of active treatment		29 (96.7)
Rescue doses		24 (80.0)
Induction doses		1 (3.7)
Hydration		1 (3.3)
**Drug regimens, *n* (%)**		
Morphine + hyoscine butyl bromide + midazolam		13 (43.3)
Morphine + hyoscine butyl bromide + midazolam + haloperidol		9 (30.0)
Morphine + hyoscine butyl bromide + haloperidol		4 (13.3)
Morphine + hyoscine butyl bromide + levomepromazine		3 (10.0)
Morphine + midazolam + haloperidol		1 (33.0)
**Drug use**		
Morphine	*n* (%)	30 (100)
	median (IQR), mg	40 (30–45)
Hyoscine butyl bromide	*n* (%)	29 (96.6)
	median (IQR), mg	40 (40–60)
Midazolam	*n* (%)	23 (76.7)
	median (IQR), mg	15 (15–30)
Haloperidol	*n* (%)	14 (46.7)
	median (IQR), mg	10 (7.50–15)
Levomepromazine	*n* (%)	3 (10)
	median (IQR), mg	75 (56–75)

## Data Availability

J.-M.R.-R., O.M.-P. and E.M. have full access to the data and are the guarantors for the data.

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
