# Peer review of "Palliative Sedation in COVID-19 End-of-Life Care. Retrospective Cohort Study"

_medicina, 2021, doi:10.3390/medicina57090873_

Round 1

Reviewer 1 Report

The manuscript entitled "End-of-life care and palliative sedation in COVID-19. Retrospective cohort study" has been reviewed. This study is valuable as it is among very few research studies which describe COVID-19 patients' characteristics and the use of palliative sedation in Spain. This manuscript is clear and well written. However, there are some major issues.

  1. The title of manuscripts implies that this study is about the "end of life care" in COVID-19 patients, however, the main focus of this study is palliative sedations and other End-of-life care factors are not investigated.
  2. As authors define in Methods "There are no specific criteria for palliative sedation used in the hospital. The reasons for palliative sedation were refractory end-stage symptoms in patient with COVID-19, decided by the patient's physician." The absence of criteria in using palliative sedation is a significant threat to the external validity of the results of this study. Since the start of the pandemic, many hospitals and professional networks created some criteria for the use of palliative sedation. These evidence based guidelines could be used in developing objective criteria for palliative care sedation in this specific hospital.
  3. In conclusions, authors highlighted the need to integrate "palliative care" based on the results of this study. Again, this study is not focused on palliative care dimensions and is just reported the use of palliative care sedation in a small sample of patients in a single hospital. Per WHO, palliative care defines as "an approach that improves the quality of life of patients and their families facing the problem associated with life-threatening illness, through the prevention and relief of suffering by means of early identification and impeccable assessment and treatment of pain and other problems, physical, psychosocial and spiritual." This study is just focused on relieving physical symptom, so generalizing the results of this study to the "Palliative Care" does not seem appropriate.

Author Response

Dear review,

We sincerely appreciate the work done by the editor and the reviewer. We believe that your comments have decisively contributed to improving the quality of our study. Thank you.

The manuscript entitled "End-of-life care and palliative sedation in COVID-19. Retrospective cohort study" has been reviewed. This study is valuable as it is among very few research studies which describe COVID-19 patients' characteristics and the use of palliative sedation in Spain. This manuscript is clear and well written. However, there are some major issues.

1.The title of manuscripts implies that this study is about the "end of life care" in COVID-19 patients, however, the main focus of this study is palliative sedations and other End-of-life care factors are not investigated.

Authors' reply, 

Following the reviewer's suggestions, we have modified the title to focus on palliative sedation rather than palliative care

Previouslly: End-of-life care and palliative sedation in COVID-19. Retrospective cohort study

Now: Palliative sedation in COVID-19 end-of-life care. Retrospective cohort study

As authors define in Methods "There are no specific criteria for palliative sedation used in the hospital. The reasons for palliative sedation were refractory end-stage symptoms in patient with COVID-19, decided by the patient's physician." The absence of criteria in using palliative sedation is a significant threat to the external validity of the results of this study. Since the start of the pandemic, many hospitals and professional networks created some criteria for the use of palliative sedation. These evidence based guidelines could be used in developing objective criteria for palliative care sedation in this specific hospital.

Authors' reply

We would like to thank the reviewer for this observation and have revised the written text in the article as suggested. We agree that it is not really the case that there were no criteria for the use of palliative sedation. The hospital has a protocol for the management of patients with COVID-19 with a section on palliative sedation. We have therefore modified that part of the text. We attach the protocol we have in Spanish in case the reviewer wants to consult it.

Previously: There are no specific criteria for palliative sedation used in the hospital. The reasons for palliative sedation were refractory end-stage symptoms in patient with COVID-19, decided by the patient's physician.

Revised text: The hospital had a protocol for managing COVID-19 patients admitted in the hospital with specific criteria for the use of palliative sedation by a palliative care team in patients with clinically worsening conditions, refractory symptoms, and/or a short-term fatal prognosis. The reasons for palliative sedation were decided by the patient's physician ac-cording this protocol.

In conclusions, authors highlighted the need to integrate "palliative care" based on the results of this study. Again, this study is not focused on palliative care dimensions and is just reported the use of palliative care sedation in a small sample of patients in a single hospital. Per WHO, palliative care defines as "an approach that improves the quality of life of patients and their families facing the problem associated with life-threatening illness, through the prevention and relief of suffering by means of early identification and impeccable assessment and treatment of pain and other problems, physical, psychosocial and spiritual." This study is just focused on relieving physical symptom, so generalizing the results of this study to the "Palliative Care" does not seem appropriate.

Authors' reply

We agree with the reviewer that the paper presents an experience specifically with palliative sedation rather than broadly with palliative care. Following the reviewer's suggestions, we have focused the discussion on palliative sedation.

Previously: These results suggest the need to integrate palliative care and pharmacological sedation in end-of-life care in inpatients with COVID-19 in acute medical wards attended by physicians not specialized in palliative care. Palliative care must be a mandatory component of multidisciplinary teams responding to COVID-19

The new conclusion is that: In conclusion, end-of-life palliative sedation in patients with COVID-19 was initiated quite late. Clinicians should anticipate palliative care needs in these patients and recognize the breathlessness, pain, and agitation/delirium that foreshadow death. These results suggest the need to correctly identify refractory symptoms at the end of life in COVID-19 patients and integrate pharmacological sedation in the care for inpatients with COVID-19 in acute medical wards attended by physicians not specialized in palliative care. Palliative sedation must be a mandatory component of multidisciplinary end-of-life approach in COVID-19.

Reviewer 2 Report

This is an interesting paper.

I do feel there is the need for more discussion about palliative sedation.  There are accepted definitions and palliative sedation implies that the prescribing doctor intended to sedate the patient for symptom management, knowing that this might continue until death.   These patients could be seen as requiring parenteral medication in doses to relieve their symptoms,  as described by "supportive medications " in lines129/130 on page 5, which led to sedation, rather than doctors intentionally providing palliative sedation with the explicit aim of sedation.  This needs to be discussed in greater depth.

Page 5 line 144 - the wording here is unfortunate - as it implies that some drugs are used more carefully than others. This should be clarified

The discussion about the role of palliative medicine specialist team involvement is confused.  It is unclear if the authors are saying that these doctors, with little training and little specialist support, were more likely to use medication in this way, or were less likely to consider sedative drugs.

There is a need for some clarity as to the main aims of the discussion.

Author Response

Dear Sir

We sincerely appreciate the work done by the editor and the reviewer. We believe that your comments have decisively contributed to improving the quality of our study. Thank you.

I do feel there is the need for more discussion about palliative sedation.  There are accepted definitions and palliative sedation implies that the prescribing doctor intended to sedate the patient for symptom management, knowing that this might continue until death.   These patients could be seen as requiring parenteral medication in doses to relieve their symptoms, as described by "supportive medications " in lines129/130 on page 5, which led to sedation, rather than doctors intentionally providing palliative sedation with the explicit aim of sedation.  This needs to be discussed in greater depth.

Author’s reply.  Following the reviewer's comment, we have changed “supportive medications” on line 120/130 for palliative sedation, writing:

Palliative sedation, prescribed to induce a state of decreased or absent awareness (unconsciousness) and relieve the burden of otherwise intractable suffering until death, was generally administered for fewer than 24 hours by the attending general physician rather than by the attending palliative physician. In this study, palliative sedation was defined solely by the use of drugs to reduce the level of consciousness when there was no response to in-tensive symptom control treatment. These were considered refractory symptoms due to the patients’ end-of-life situation.

Page 5 line 144 - the wording here is unfortunate - as it implies that some drugs are used more carefully than others. This should be clarified

Author’s reply, Following the reviewer's comment we have rewritten this part of the discussion. It now reads:

“ … less than in other studies where medical staff or advising physicians from palliative care units prescribed the palliative sedation for a longer period for symptom management.”

The discussion about the role of palliative medicine specialist team involvement is confused.  It is unclear if the authors are saying that these doctors, with little training and little specialist support, were more likely to use medication in this way, or were less likely to consider sedative drugs.

There is a need for some clarity as to the main aims of the discussion.

Following the reviewer's comment, we have included in the discussion several items about that with several new references.

New paragraph

A recent systematic review of the clinical aspects of palliative sedation in the non-COVID-19 population showed that the decision-making process was led mainly by the palliative care team, and in some cases by the attending palliative care physician [16]. Another study, focused on general practitioners, highlighted the importance of working with palliative care professionals due to GPs’ limited experience in palliative sedation [17]. In the case of COVID-19, an acute infectious disease with high mortality, several hospitals have developed guidelines that could be used for attending physicians who are not specialists in palliative care, with objective criteria for palliative care sedation in these situations [18].

The new references are:

16.Arantzamendi M, Belar A, Payne S, Rijpstra M, Preston N, Menten J, Van der Elst M, Radbruch L, Hasselaar J, Centeno C. Clinical Aspects of Palliative Sedation in Prospective Studies. A Systematic Review. J Pain Symptom Manage. 2021;61:831-844.e10. doi: 10.1016/j.jpainsymman.2020.09.022.

17.Pype P, Teuwen I, Mertens F, Sercu M, De Sutter A. Suboptimal palliative sedation in primary care: an exploration. Acta Clin Belg 2018;73:21-28. doi: 10.1080/17843286.2017.1331783

18.Mitchell S, Maynard V, Lyons V, Jones N, Gardiner C. The role and response of primary healthcare services in the delivery of palliative care in epidemics and pandemics: A rapid review to inform practice and service delivery during the COVID-19 pandemic. Palliat Med. 2020;34:1182-1192
